# Directed evolution of aminoacyl-tRNA synthetases through in vivo hypermutation

Yuichi Furuhata[1,2,3], Gordon Rix[3,4], James A. Van Deventer [5,6] &
Chang C. Liu [1,3,4,7] ✉

Genetic code expansion (GCE) is a critical approach to the site-specific incorporation of non-canonical amino acids (ncAAs) into proteins. Central to GCE is the development of orthogonal aminoacyl-tRNA synthetase (aaRS)/ tRNA pairs wherein engineered aaRSs recognize chosen ncAAs and charge them onto tRNAs that decode blank codons (e.g., the amber stop codon). However, evolving new aaRS/tRNA pairs traditionally relies on a labor-intensive process that often yields aaRSs with suboptimal ncAA incorporation efficiencies. Here, we present an OrthoRep-mediated strategy for aaRS evolution, which we demonstrate in 8 independent aaRS evolution campaigns, yielding multiple aaRSs that incorporate an overall range of 13 ncAAs tested. Some evolved systems enable ncAA-dependent translation at single amber codons with similar efficiency as natural translation at sense codons. Additionally, we discover an aaRS that regulated its own expression to enhance ncAA dependency. These findings demonstrate the potential of OrthoRep-driven aaRS evolution platforms to advance the field of GCE.

Over the past 30 years, genetic code expansion (GCE) has matured into a versatile technology for protein science and engineering by enabling the direct site-specific incorporation of non-canonical amino acids (ncAAs, also known as uAAs for unnatural amino acids or nsAAs for nonstandard amino acids) into proteins in vivo[1]. GCE has been implemented in a wide range of organisms – from workhorse microbes (e.g., *Escherichia coli* and *Saccharomyces cerevisiae*) to mammalian cell lines and animals[2,3] – for >200 distinct ncAAs[4,5]. Such breadth and flexibility in augmenting the chemical functionality of proteins has resulted in numerous advances in basic protein biology, such as the use of ncAAs as probes of structure and dynamics[6–8]; therapeutic protein engineering, such as the generation of defined antibody-drug conjugates[9–13]; enzyme engineering, such as the installation of non-canonical nucleophilic or photosensitizer amino acids to expand catalytic scope[14–19]; imaging, such as the addition of fluorogenic amino acids to monitor proteins and complexes in vivo[20–24]; chemical biology,

such as the direct expression of modified proteins that recapitulate specific posttranslational modification patterns to study their biological roles[25–31]; and cell biology, such as the use of reactive ncAAs to map interactomes[32,33]. In addition to imbuing proteins with novel chemistries, GCE has played a key role in synthetic genetics, such as the development of genetic firewalls where certain essential proteins are designed to require an ncAA, thus enforcing ncAA-dependent growth and ncAA-based biocontainment of the host organism[34]. GCE has also been used to establish links in synthetic gene circuits where GCE components act as biosensors of ncAA production[35] or ncAA-dependent switches in genetic programs and logical operations[36,37]. To support the continued growth of this rich ecosystem, we present a streamlined approach for the development of efficient GCE systems.

At the heart of GCE is the establishment of an orthogonal aminoacyl-tRNA synthetase (aaRS)/tRNA pair[2,3], sometimes called an orthogonal translation system. The orthogonal aaRS should

[1]Department of Biomedical Engineering, University of California, Irvine, CA, USA. [2]Molecular Biosystems Research Institute, National Institute of Advanced Industrial Science and Technology (AIST), Tsukuba, Ibaraki, Japan. [3]Center for Synthetic Biology, University of California, Irvine, CA, USA. [4]Department of Molecular Biology and Biochemistry, University of California, Irvine, CA, USA. [5]Department of Chemical and Biological Engineering, Tufts University, Medford, MA, USA. [6]Department of Biomedical Engineering, Tufts University, Medford, MA, USA. [7]Department of Chemistry, University of California, Irvine, CA, USA. ✉e-mail: ccl@uci.edu

specifically aminoacylate its cognate orthogonal tRNA (and not endogenous host tRNAs) with an ncAA (and not canonical AAs (cAAs)). The orthogonal tRNA should be aminoacylated only by the orthogonal aaRS (and not host aaRSs). The tRNA should specifically recognize a blank codon (codon$_{BL}$) – most commonly the amber stop codon – that is not already assigned to a canonical amino acid (cAA)[2,3]. If these conditions are met, an organism expressing the orthogonal aaRS/tRNA pair will then decode the codon$_{BL}$ with an ncAA such that any coding sequence containing a reading frame codon$_{BL}$ templates the cotranslational incorporation of the ncAA into the resulting protein. This basic architecture for GCE, solidified in the early 2000s[1,38], continues to be developed on multiple distinct and creative fronts, including the identification of suitable aaRS/tRNA pairs that are orthogonal in new host organisms[3], the expansion of available codon$_{BL}$'s beyond amber through genome synthesis[39], the discovery and engineering of mutually orthogonal aaRS/tRNA pairs for the incorporation of two or more distinct ncAAs into the same protein[40], and the engineering of ribosomes and other components of protein translation to better interface with ncAAs[41]. Despite these myriad ways the basic GCE architecture has been extended, all GCE approaches still share the need for aaRSs that recognize an ever-growing scope of ncAAs with high efficiency, which has remained a persistent challenge since the beginnings of the GCE field. Notably, current GCE systems often exhibit low efficiencies of ncAA incorporation into proteins[42], even for simple applications involving the specification of only one ncAA into one protein. This is especially true in yeast where GCE efforts have historically lagged[43,44] even though the opportunities are ample – for example, yeast surface display discovery of antibodies that use reactive ncAAs to crosslink their target[45]. Additionally, yeast can be an ideal organism to evolve/engineer aaRSs, since its endogenous translational machinery is orthogonal to a wide range of bacterial and archaeal aaRS/tRNAs that are popular in the GCE field[3]. Moreover, the creation of new or improved GCE machinery currently relies on specialized and labor-intensive protein engineering campaigns[3], creating a bottleneck in the scaling of GCE to more organisms, ncAAs, and applications.

We report an effective approach to aaRS engineering where aaRSs are encoded on an orthogonal error-prone DNA replication system (OrthoRep)[46,47] in the yeast *S. cerevisiae*, resulting in populations that autonomously diversify aaRSs. When subjected to cycles of positive selection for the productive expression of an amber codon-containing reporter gene in the presence of ncAAs and negative selection for the lack of reporter expression in the absence of ncAAs, we observe the rapid evolution of aaRSs exhibiting high efficiencies of ncAA-dependent protein translation, exceeding those of previous efforts that involved traditional directed evolution of aaRSs for yeast GCE[48–50] and reaching levels of expression that match translation with only cAAs in certain cases. We demonstrate OrthoRep-driven evolution for several orthogonal aaRS/tRNA pairs against several different ncAAs, including aaRSs based on the versatile *Methanomethylophilus alvus* pyrolysyl-tRNA synthetase (PylRS)/tRNA$^{Pyl}$ pair known to function in a wide range of organisms[51–53]. We also report an aaRS that evolved to control its own expression by ncAAs, exemplifying the emergence of an autoregulatory mechanism that minimizes translational leak in the absence of ncAAs. We attribute the high efficiency of our evolved aaRSs and the surprise generation of a self-regulating aaRS to the rapid, scalable, and open-ended nature of OrthoRep-driven aaRS evolution and suggest its broad potential role in the GCE field going forward.

## Results and discussions
### Evolution of aaRSs with OrthoRep
To develop an OrthoRep-driven aaRS evolution platform, we encoded aaRSs on the hypermutating orthogonal plasmid of OrthoRep (p1) and adopted a ratiometric RXG reporter[54], where RFP and GFP are connected by a linker containing an amber stop codon, as the selection

system for evolution (Fig. 1a). We initially attempted a growth-based selection where the aminoacylation of an orthogonal amber suppressor tRNA in the presence of an ncAA would increase the expression of a URA3 gene containing an amber stop codon. Positive selection for ncAA-dependent translation of the amber codon would be achieved in the presence of ncAA and absence of exogenously added uracil, and negative selection would be achieved in the absence of ncAA and presence of 5-fluoroorotic acid (5-FOA) and uracil. However, this approach was not successful, presumably due to large copy number variation of the *CEN/ARS* plasmid on which we encoded the URA3 gene. We anecdotally observed a substantial increase in the *CEN/ARS* plasmid's copy number during positive selection phases, suggesting that cells evolved to express sufficient URA3 protein by increasing background readthrough of its amber stop codon with elevated URA3 gene dosage rather than through enhanced aminoacylation of the orthogonal amber suppressor tRNA by the aaRS. To avoid this challenge, we turned to a ratiometric RXG reporter[54] to guide aaRS evolution. This RXG reporter configuration allowed us to directly select cells based on the readthrough ratio facilitated by an aminoacylated orthogonal tRNA rather than rely on the absolute amount of reporter protein expression. We were specifically interested in evolving aaRSs with a high level of amber codon readthrough in the presence versus the absence of each target ncAA. In other words, we were interested in a high $\frac{GFP}{RFP}$ fluorescence ratio in the presence of ncAAs and a low $\frac{GFP}{RFP}$ fluorescence ratio in the absence of ncAAs. To normalize fluorescence measurements, we determined GFP and RFP fluorescence of an RYG reporter that contained a sense codon specifying Y instead of an amber stop codon, which we assume results in the complete translation of full-length RFP-GFP fusion protein. We refer to the quantity $\left(\frac{GFP}{RFP}\ \text{fluorescence for RXG}\right)/\left(\frac{GFP}{RFP}\ \text{fluorescence for RYG}\right)$ as the relative readthrough efficiency (RRE)[54,55] and our goal was to evolve aaRSs that have high RREs in the presence of specific ncAAs and low RREs in their absence. For the OrthoRep-driven aaRS evolution strain, we started from *S. cerevisiae* strain LLYSS4, which has deletions of LEU2 and TRP1, a split LEU2 landing pad p1[47], and a WT orthogonal DNA polymerase (DNAP) expression cassette at the CAN1 locus[47]. We integrated an aaRS gene onto the split LEU2 landing pad, transformed a *CEN/ARS* plasmid carrying an RXG reporter and the corresponding orthogonal amber suppressor tRNA expression unit, and replaced the WT orthogonal DNAP cassette with an error-prone orthogonal DNAP (epDNAP) expression cassette (Fig. 1a). In the resulting strain, the chosen epDNAP continuously replicates the aaRS sequence at a high mutation rate (*i.e.*, 10$^{-5}$ substitutions per base (s.p.b.) for BB-Tv[47], the primary epDNAP used in this study) while sparing the host genome from elevated mutagenesis.

To evolve aaRSs that aminoacylate their cognate amber suppressor tRNA with various ncAAs, fluorescence-activated cell sorting (FACS)-based evolution cycles were performed. Cells were grown for ~35 generations at 30 °C after the introduction of epDNAP before cycles began. Then, each cycle consisted of the following steps: (1) induction of RXG expression with galactose in the presence or absence of ncAA for 48 h at 30 °C, which corresponds to only ~3 generations, since cells grow slowly under induction; 2a) a positive selection sort on ~10,000,000 cells in which the 0.05% of cells with the highest GFP to RFP ratio in the presence of ncAA were collected, or 2b) a negative selection sort on ~1,000,000 cells in which the 5% of cells with the lowest GFP to RFP ratio in the absence of ncAA were collected; and 3) growth of sorted cells to saturation at 30 °C, corresponding to ~13–15 generations depending on the amount of cells collected (Fig. 1a). The ncAAs used in this study are shown in Fig. 1b. A negative selection sort was performed every 2 to 3 cycles to remove cells that could express GFP in the absence of ncAAs, such as ones that may have evolved aaRS variants that aminoacylate the amber suppressor tRNA with cAAs. Because genes encoding aaRSs autonomously diversify during culturing, cycles of this process led to aaRS variants with improved

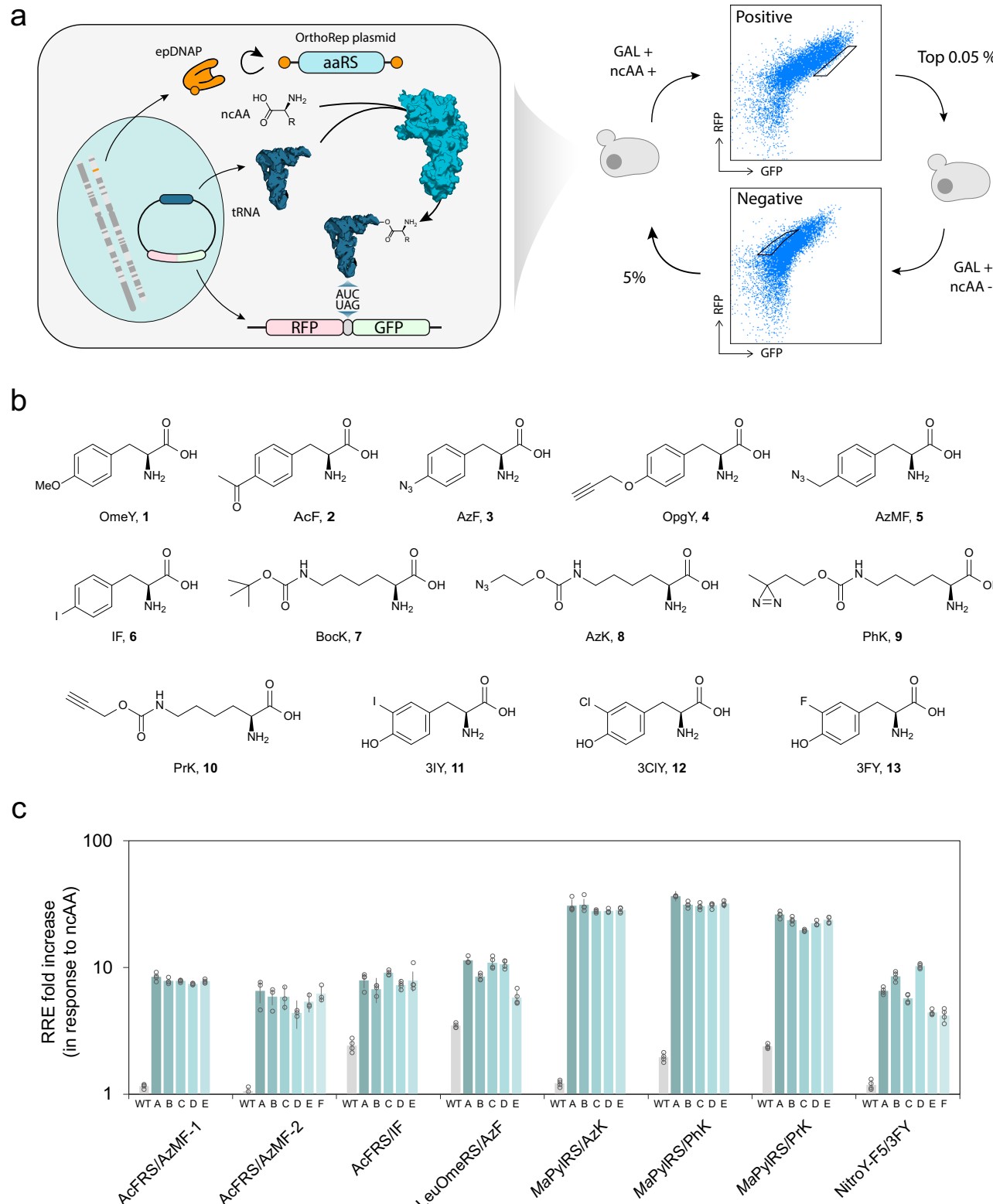

**Fig. 1 | Directed evolution of aaRSs with OrthoRep. a** Schematic for a directed evolution approach to evolve the activity of aaRSs of interest for ncAAs of interest. **b** ncAAs used in this study. **c** ncAA incorporation by individual evolved aaRS variants with individual target ncAA. RRE with target ncAA (10 mM for 3FY, 1 mM for the other ncAAs) was divided by that without ncAA. Each condition was measured in technical triplicates for the AcFRS/AzMF-2 variants and quadruplicates for the other variants, and the mean ± one standard deviation (error bars) is shown. Source data are provided as a Source Data file.

activity and selectively for ncAAs over time. After the last cycle, aaRS sequences on p1 were isolated via polymerase chain reaction (PCR), subcloned into a *CEN/ARS* plasmid under the RPL18b promoter and transformed into a fresh yeast strain carrying the same reporter *CEN/ARS* plasmid for a final negative selection and positive selection sort where hypermutation was absent. After this final sort, approximately 40 clones were randomly picked and characterized. The top 5–6 clones exhibiting the highest RREs in the presence of their target ncAAs and low RREs in the absence of their target ncAAs were chosen for further testing.

We selected 4 aaRSs as parents to evolve: AcFRS, LeuOmeRS, *Ma*PylRS, and NitroY-F5. These parents are, respectively, an engineered TyrRS from *E. coli*[49], an engineered LeuRS from *E. coli*[56], the wild-type PylRS from *Methanomethylophilus alvus*[52,57], and an engineered *M. alvus* PylRS[58]. In total, we conducted 8 independent evolution campaigns (designated as parental aaRS/ncAA target followed by a number if more than one campaign on the same components was done). The 8 evolution campaigns are therefore designated as AcFRS/AzMF-1, AcFRS/AzMF-2, AcFRS/IF, LeuOmeRS/AzF, *Ma*PylRS/AzK, *Ma*PylRS/PhK, *Ma*PylRS/PrK, and NitroY-F5/3FY throughout the manuscript. The detailed conditions for the evolution campaigns including ncAA concentrations are shown in Supplementary Data 1. In all campaigns, the parental aaRS had weak or no detectable read-through activity in the presence of its target ncAA. Throughout evolution campaigns, we observed substantial increases in ncAA-dependent RREs and isolated 5–6 individual clones (designated as a letter appended to the evolution campaign from which they arose) that exhibited greatly improved RREs for their target ncAAs compared to their parental aaRSs in all cases (Fig. 1c). Due to the ratiometric nature of RREs, we note that RREs from this study can be compared with other work reporting the same (see Methods).

## Characteristics of evolved *E. coli* aaRSs

To characterize the performance of evolved *E. coli* aaRSs, we sequenced top-performing clones and measured their RREs in the presence and absence of 6 tyrosine or phenylalanine analogs including target ncAAs (Fig. 2a, Supplementary Data 2, 3). The parental AcFRS was previously shown to be promiscuous, exhibiting activity for various tyrosine or phenylalanine analogs including OmeY (**1**), AcF (**2**), AzF (**3**), and OpgY (**4**). However, it had minimal activity for AzMF (**5**) and IF (**6**) (Fig. 2a). The evolution campaigns AcFRS/AzMF-1, AcFRS/AzMF-2, and AcFRS/IF, improved the activities of AcFRS for those two ncAAs. Interestingly, all the top-performing variants from the AcFRS/AzMF-1 and AcFRS/AzMF-2 campaigns also exhibited improved activity for IF (**6**), despite not being selected for IF (**6**), and vice-versa. Moreover, they all maintained decent activities for the other ncAAs. This suggests that the OrthoRep-driven evolution campaigns expanded AcFRS's activity in a way that maintained or increased its promiscuity for a set of ncAAs.

While the evolved AcFRS variants showed a similar substrate preference for all the ncAAs we tested, the mutations they acquired varied considerably among the 3 campaigns. The top 5 variants from AcFRS/AzMF-1 contained between 2 and 5 amino acid mutations, including G182S in the substrate binding pocket, which was shared among all the variants. The G182S mutation was previously identified for IF incorporation[49], consistent with its prevalence across evolved variants here. Our top variants exhibited higher RREs for AzMF (**5**), ranging from 0.327 to 0.457, while WT AcFRS had a low RRE of 0.047. Although clone AcFRS/AzMF-1-D exhibited the highest RRE, 0.457, with AzMF (**5**), it also displayed the highest RRE, 0.062, without ncAA. Among the top 5 variants, AcFRS/AzMF-1-A showed the lowest RRE, 0.039, without ncAA and the highest RRE fold-increase, 8.44-fold, in the presence versus absence of AzMF (**5**). AcFRS/AzMF-1-A harbors 4 additional mutations, including L71M in the substrate binding pocket (Fig. 2a, b). Furthermore, AcFRS/AzMF-1-A demonstrated a higher RRE

fold-increase in the presence of other ncAAs as well (Supplementary Data 3), owing to its exceptionally low RRE in the absence of ncAAs.

In contrast to the AcFRS/AzMF-1 evolution campaign, the top 6 variants from the AcFRS/AzMF-2 campaign contained only 1 to 3 amino acid mutations, including W260L shared among all variants. There were no other amino acid mutations in 4 out of the 6 variants, indicating that W260L played a pivotal role (Fig. 2a). W260L appeared to generally increase the aminoacylation activity of AcFRS, including elevation of RRE in the absence of ncAAs. For example, AcFRS/AzMF-2-A exhibited an RRE of 0.493 with AzMF (**5**) and an RRE of 0.076 without ncAA, whereas the without-ncAA RRE of WT AcFRS was 0.031. Interestingly, W260 is located relatively distal to the substrate binding pocket (Fig. 2c). This indicates the value of using OrthoRep for aaRS evolution, as W260 would likely be omitted for diversification in traditional aaRS evolution approaches based on saturation mutagenesis libraries that prioritize the active site. Intriguingly, we observed that 2 of the 6 variants with W260L also had an amber codon in the evolved aaRS gene itself. Given that W260L increased background aminoacylation activity in the absence of ncAA but that we also negatively selected against such background activity during evolution, the amber codon mutation within the aaRS sequence likely enriched by lowering background activity through ncAA-dependent autoregulation of aaRS expression itself, which we further investigated and validated (see below).

The top 5 variants in the AcFRS/IF campaign contained between 1 and 3 amino acid mutations, including V101A shared among all variants (Fig. 2a). These evolved aaRSs exhibited higher RREs for IF (**6**) ranging from 0.283 to 0.318, compared to 0.078 for WT AcFRS. AcFRS/IF-C, the top performing variant, exhibited an RRE of 0.302 with IF (**6**) and 0.033 without ncAA, resulting in a 9.09-fold ncAA-dependent RRE increase, while WT AcFRS only exhibited a 2.42-fold ncAA-dependent RRE increase. AcFRS/IF-C had only one mutation, the V101A, indicating its crucial role in conferring IF (**6**) activity. Although V101 is proximal to the substrate binding pocket (Fig. 2d), there is no direct interaction between V101 and the substrate in the crystal structure of AcFRS (PDB: 6HB5). While we observed a variety of mutations in the 3 different AcFRS evolution campaigns described, all rendered AcFRS more active for both AzMF (**5**) and IF (**6**).

LeuOmeRS was previously shown to be a promiscuous aaRS that exhibits robust activities for OmeY (**1**), OpgY (**4**), AzMF (**5**), and IF (**6**), while displaying poor activities for AcF (**2**) and AzF (**3**)[54]. The top 5 variants from the LeuOmeRS/AzF evolution campaign exhibited improved activity for AzF (**3**) with RREs ranging from 0.269 to 0.484. Notably, the evolved variants exhibited reduced activity for almost all other ncAAs tested, constituting evolutionary outcomes with increased AzF-specificity. This contrasts with AcFRS evolution campaigns that resulted in high activity generalists for the ncAAs profiled (Fig. 2a). The top 5 variants contained between 5 and 11 amino acid mutations, including L40F and A527V, which were shared among all variants. Additionally, G379S and Q831R, distal to the substrate binding pocket (~44 Å and ~61 Å, respectively), were also shared among 3 out of 5 variants, including LeuOmeRS/AzF-A, the top-performing variant (Fig. 2e). LeuOmeRS/AzF-A exhibited an RRE of 0.484 with AzF (**3**) and 0.042 without ncAA, resulting in an 11.40-fold ncAA-dependent RRE increase. LeuOmeRS/AzF-D had a unique sequence and exhibited the lowest RRE value of 0.029 without ncAA. It seems likely that the mutations specific to LeuOmeRS/AzF-D (A126T, D173, P455L, and M494I) contributed to lowering background activity in the absence of ncAA. Overall, we identified a cluster of mutations, including the core mutations L40F and A527V, that improved activity for AzF (**3**).

## Characteristics of evolved *M. alvus* aaRSs

We sequenced the top performing *Ma*PylRS and NitroY-F5 variants and measured their RREs with 4 lysine or 3 tyrosine analogs (Fig. 3a, Supplementary Data 2, 4). The parental *Ma*PylRS is highly active for BocK

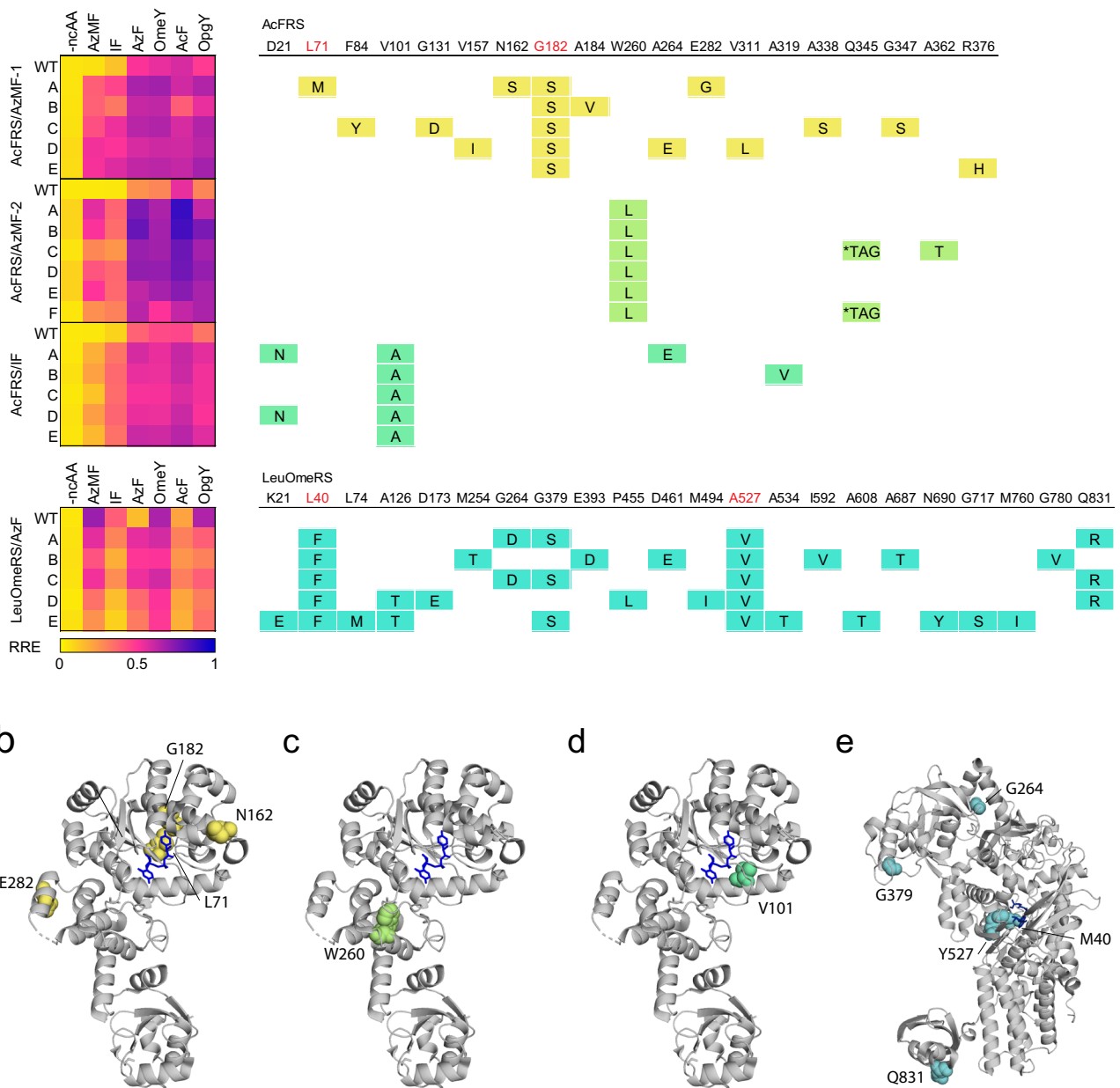

**Fig. 2 | Evolved *E. coli* aaRSs and their preferences for ncAAs. a** Heatmap of RRE values with 6 tyrosine or phenylalanine analogs for the top aaRS clones sampled from the evolution experiment and table depicting their mutations. Fold changes in RRE values (with ncAA/without ncAA) are shown in Supplementary Data 3. Residues targeted for saturation mutagenesis in a previous study[48] are shown in red. Crystal structure of *Ec*TyrRS (PDB 6HB5) (**b**–**d**) and *Ec*LeuRS (PDB 4CQN) (**e**) with mutated residues of the top performing clone from each evolution campaign highlighted in colors corresponding to the table (**a**). Amino acid substrates are shown as blue sticks. *Ec*TyrRS and *Ec*LeuRS are the parent aaRSs of AcFRS and LeuOmeRS, respectively. Source data are provided as a Source Data file.

(**7**) but exhibits minimal activity for AzK (**8**), PhK (**9**), and PrK (**10**) (Fig. 3a). The top 5 variants from the *Ma*PylRS/AzK campaign all exhibited substantial improvements in their activities for AzK (**8**) and PhK (**9**), moderate improvements in their activities for PrK (**10**), and decreased activity for BocK (**7**), suggesting that evolution was dominated by both activity and selectivity increases. For instance, *Ma*PylRS/AzK-B, the top performing variant, exhibited RREs of 0.919, 0.967, 0.352, and 0.452 for AzK (**8**), PhK (**9**), PrK (**10**), and BocK (**7**), while the corresponding RREs for WT *Ma*PylRS were 0.036, 0.056, 0.070, and 0.710, respectively (Fig. 3a, b). We note that for AzK (**8**) and PhK (**9**), the RREs for evolved aaRSs approached 1.0, suggesting that translation of amber codon-containing genes occurred at similar efficiency as the

natural translation of genes with cAAs specified by sense codons. The ncAA-dependent RRE increase for *Ma*PylRS/AzK-B with AzK (**8**) was 31.26-fold, while that for WT *Ma*PylRS was 1.22-fold, reflecting a substantial improvement in the selective incorporation of AzK (**8**). The top 5 variants contained 3 to 5 amino acid mutations, including M129V in the substrate binding pocket, which was shared among all variants. Additionally, mutations at G17 and I210 were shared among 3 variants each. Interestingly, G17S, G17D, and I210T were also observed in the *Ma*PylRS/PrK campaign. Indeed, *Ma*PylRS/AzK-B, *Ma*PylRS/AzK-D, and *Ma*PylRS/AzK-E, which harbored 1 or 2 of these mutations, exhibited higher RREs compared to *Ma*PylRS/AzK-A and *Ma*PylRS/AzK-C for PrK (**10**), indicating their contribution to PrK (**10**) activity (Fig. 3a). These

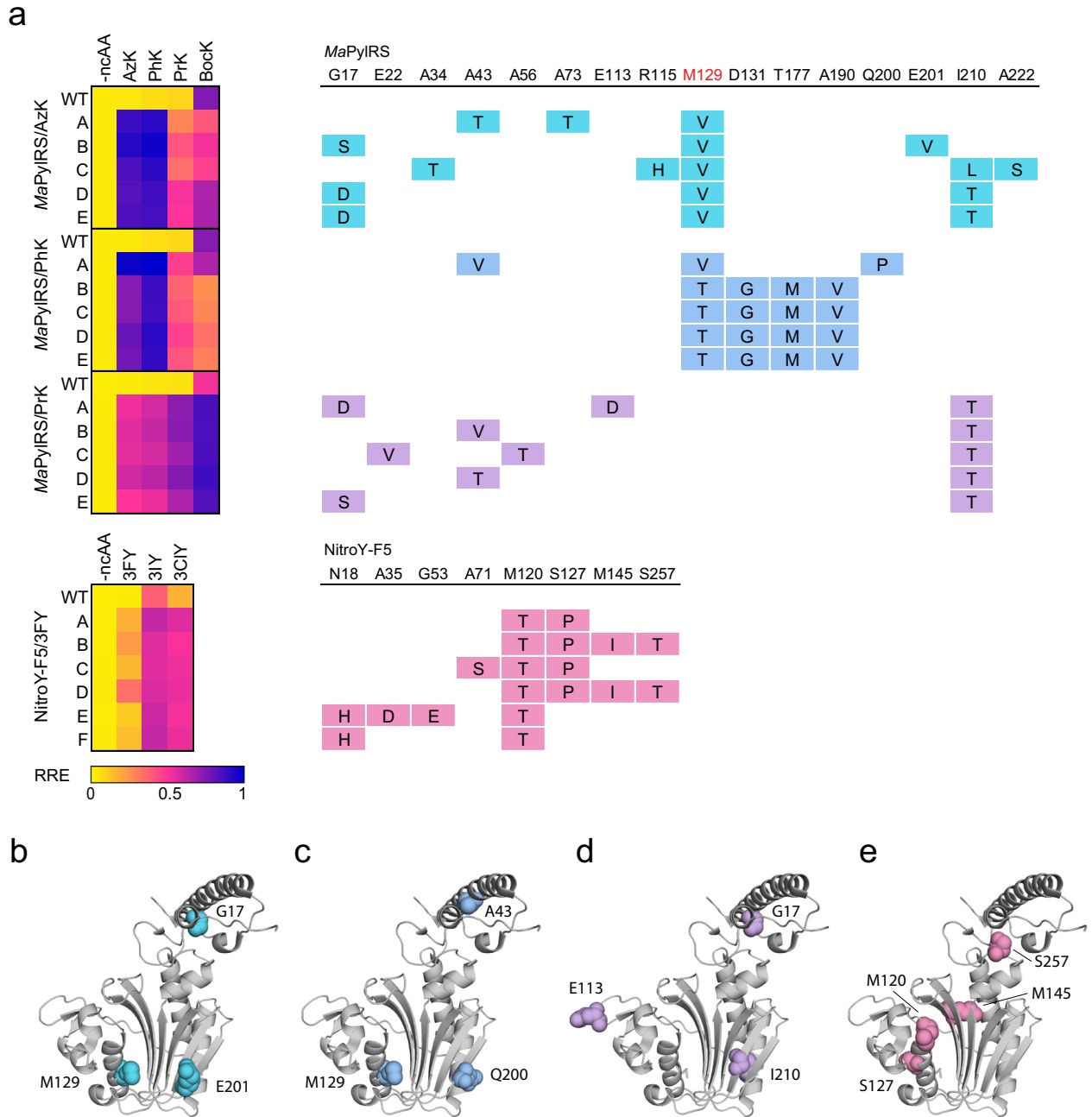

**Fig. 3 | Evolved *M. alvus* aaRSs and their preferences for ncAAs. a** Heatmap of RRE values with 4 lysine or 3 tyrosine analogs for evolved aaRS variants and table depicting their mutations. Fold changes in RRE values (with ncAA / without ncAA) are shown in Supplementary Data 4. A residue targeted for saturation mutagenesis in a previous study[58] is shown in red. **b–e** Crystal structure of *Ma*PylRS (PDB 6JP2) with mutated residues of the top performing clone from each evolution campaign highlighted in colors corresponding to the table (**a**). *Ma*PylRS is the parent aaRS of NitroY-F5. Source data are provided as a Source Data file.

mutations might be involved in interactions with the tRNA, since G17 is in the tRNA binding domain of *Ma*PylRS.

The top 5 variants from the *Ma*PylRS/PhK campaign exhibited a similar ncAA preference to that of the *Ma*PylRS/AzK variants – they substantially improved activity for both AzK (**8**) and PhK (**9**) while decreasing activity for BocK (**7**) (Fig. 3a). *Ma*PylRS/PhK-A, the top performing variant, exhibited RREs of 0.973, 1.014, 0.400, and 0.608 for AzK (**8**), PhK (**9**), PrK (**10**), and BocK (**7**), respectively. These observations suggest a strong correlation between *Ma*PylRS activity for AzK (**8**) and PhK (**9**). Evolved clones contained between 3 and 4 amino acid mutations, including M129V/T in the substrate binding pocket shared among all the variants. Although 4 out of the 5 variants shared an identical sequence, *Ma*PylRS/PhK-A, featuring a unique

sequence with A43V, M129V, and Q200P, demonstrated the best performance for all tested ncAAs (Fig. 3a, c). While several mutations at M129 have been reported in previous studies[52,59,60], the M129V and M129T mutations identified in this study have not been reported to the best of our knowledge.

In contrast, the top 5 variants from the *Ma*PylRS/PrK campaign, which displayed significant improvements in PrK (**10**) activity, also exhibited enhanced activity for BocK (**7**) (Fig. 3a). *Ma*PylRS/PrK-A, the top performing variant, exhibited RREs of 0.464, 0.531, 0.683, and 0.834 for AzK (**8**), PhK (**9**), PrK (**10**), and BocK (**7**), respectively. The ncAA-dependent RRE increase for *Ma*PylRS/PrK-A with PrK (**10**) was 26.13-fold, while that of WT *Ma*PylRS was only 2.39-fold. The top 5 variants contained between 2 and 3 amino acid mutations, including

I210T, which was shared among all variants. I210 is a previously unreported residue affecting ncAA selectivity of *Ma*PylRS and is located proximal to the substrate binding pocket (Fig. 3d). In addition, mutations at G17 and A43 were also shared among 2 variants each. Given their locations, these might be beneficial for interacting with the tRNA.

NitroY-F5 is an engineered *Ma*PylRS capable of incorporating 3IY (**11**) and 3ClY (**12**) at high efficiency[58]. However, NitroY-F5 did not exhibit activity for 3FY (**13**) (Fig. 3a). The top 5 variants from the NitroY-F5/3FY campaign showed substantially improved activity for 3FY (**13**) as well as 3IY (**11**) and 3ClY (**12**) (Fig. 3a). These variants contained 2 to 4 mutations, including M120T, which was shared among all 6 variants. Additionally, NitroY-F5/3FY-B and NitroY-F5/3FY-D, the top performing variants, also contained S127P, M145I, and S257T (Fig. 3e). Overall, we identified several mutations, both proximal and distal to the substrate binding pocket of *Ma*PylRS, that modulated aminoacylation activity and substrate scope.

Given the high RREs that some evolved *Ma*PylRSs achieved, we wished to test whether they could induce a fitness cost through global suppression of amber stop codons for host genes. We performed growth rate assay for a yeast strain expressing *Ma*PylRS/PhK-A, which exhibited the highest RRE of 1.014 in the presence of 1 mM PhK (**9**) (Fig. 3a), alongside an otherwise identical strain containing an empty vector instead of the *Ma*PylRS/PhK-A expression plasmid. The strain expressing *Ma*PylRS/PhK-A only showed a slightly decreased growth rate (-5% reduction per hour) in the presence of PhK (**9**) but could nonetheless grow and saturate without notable practical limitations (Supplementary Fig. 1). For ncAAs that had lower RREs with *Ma*PylRS/PhK-A, a lower reduction in growth rate was observed. Therefore, the high RREs of evolved *Ma*PylRSs are suitable for application.

## ncAA response functions and fidelity of evolved aaRSs

To characterize the ncAA concentration-dependence of aaRSs, we obtained in vivo ncAA response curves for the best-performing variant from each evolution campaign (Fig. 4a–g). Strains were induced with galactose for 48 h at 30 °C in a range of ncAA concentrations up to 10 mM, and RRE values were determined for all concentrations. For comparison, the parental aaRSs were tested alongside the evolved variants. All variants evolved in this study outperformed their parental aaRSs in their responses to the corresponding ncAAs (Fig. 4a–g). We also calculated the limit of detection (LOD) concentration for each aaRS, which we defined as the ncAA concentration that gave an RRE three standard deviations above the RRE without ncAA (Fig. 4h). Remarkably, all the evolved variants in this study surpassed their parent aaRSs, with LOD concentrations 29- to 8500-fold lower than their parents. Notably, *Ma*PylRS/PhK-A showed detectable ncAA-dependent readthrough at 9 nM of PhK (**10**) in the culture, which is over four orders of magnitude lower than cAA concentrations used in synthetic media (~ 0.3–1.0 mM), whereas the LOD concentration of *Ma*PylRS for PhK (**10**) was 44 μM.

To assess the fidelity of ncAA incorporation by the OrthoRep-evolved aaRSs, we expressed sfGFP containing an amber codon at position 150 using the best-performing variants from each campaign in the presence of cognate ncAAs. The resulting sfGFPs were purified and analyzed by whole protein mass spectrometry. Intact protein mass spectrometry revealed the expected mass corresponding to incorporation of corresponding ncAA at position N150 by all RSs except NitroY-F5/3FY-D, where we could not obtain enough sfGFP-150TAG protein due to its relatively low RRE with 3FY (**13**) and the toxicity of 3FY (**13**) at 10 mM (Supplementary Fig. 2). These results indicate that OrthoRep-driven aaRS evolution can produce aaRSs with exceptional sensitivity and fidelity for their corresponding ncAAs, which are useful not only for GCE but also for the repurposing of aaRSs as biosensors of ncAA production.

## Activity of evolved aaRSs in *E. coli*

In the OrthoRep-driven evolution campaigns, we identified many mutations, both proximal and distal to the ncAA binding pocket of aaRS, that modulated aminoacylation activity and substrate scope in *S. cerevisiae*. To test whether these mutations are also effective in other organisms, we evaluated the best-performing *Ma*PylRS and NitroY-F5 variants (*Ma*PylRS/AzK-B, *Ma*PylRS/PhK-A, *Ma*PylRS/PrK-A, and NitroY-F5/3FY-D) for their ability to incorporate each corresponding ncAA in *E. coli*. All aaRSs were cloned into the pUltra vector containing an amber suppressor tRNA (*Mat*RNA$^{Pyl}_{CUA}$) and co-trasformed into TOP10 *E. coli* cells together with a GFP reporter plasmid containing TAG at position 3 (Supplementary Fig. 3a). aaRS activity was evaluated by GFP fluorescence in the presence and absence of ncAA. All 4 best-performing variants produced GFP fluorescence in the presence of corresponding ncAA but not in its absence (Supplementary Fig. 3b). *Ma*PylRS/AzK-B and *Ma*PylRS/PhK-A, which exhibited high RREs close to 1.0 in *S. cerevisiae* (Fig. 3a), also showed comparable fluorescence with target ncAA to that of WT GFP in *E. coli* (Supplementary Fig. 3a). In contrast, *Ma*PylRS/PrK-A exhibited low activity with 1 mM PrK (**10**) in *E. coli* despite its high RRE of 0.683 in *S. cerevisiae*. We suspect that this is caused by differences in ncAA transport, and therefore different intracellular ncAA concentrations, across different organisms. Consistent with this hypothesis, *Ma*PylRS/PrK-A had lower RREs with PrK (**10**) in a concentration range of 0.01–1 mM compared to *Ma*PylRS/AzK-B and *Ma*PylRS/PhK-A with corresponding ncAAs (Fig. 4). In addition, *Ma*PylRS/PrK-A exhibited higher activities in *E. coli* when we increased the concentration of PrK (**10**) (Supplementary Fig. 3c). Overall, these results indicate that the aaRS variants obtained with OrthoRep are effective in organisms beyond yeast, demonstrating the broader utility of OrthoRep-driven aaRS evolution for GCE.

## Surprise evolution of autoregulation

In the AcFRS/AzMF-2 campaign, we observed a CAG to TAG mutation at Q345 in 2 out of 6 top-performing evolved aaRS variants (Figs. 2a, 5a). Naively, this premature amber codon at Q345 should compromise the aaRS's expression. However, since the aaRS's evolved function was to suppress the amber codon with AzMF (**5**), we inferred that the TAG-containing aaRS gene could itself be fully translated. Moreover, we hypothesized that the TAG amber codon in the aaRS coding region was adaptive because it would reduce background aaRS activity in the absence of ncAAs. In essence, we reasoned that what had evolved was an autoregulatory positive feedback loop controlling the expression of the aaRS in response to the AzMF (**5**) the aaRS evolved to recognize it (Fig. 5b).

To test this hypothesis, we investigated the behavior of our TAG-containing aaRS variant (AcFRS/AzMF-2-F) alongside an otherwise identical mutant lacking the TAG (AcFRS/AzMF-2-A). We observed that compared to AcFRS/AzMF-2-A, which was the other top variant that the AcFRS/AzMF-2 campaign yielded (Fig. 2a), AcFRS/AzMF-2-F indeed exhibited a greater ncAA-dependent RRE fold-increase for almost all ncAAs for which AcFRS/AzMF-2-F had promiscuous activity (Figs. 2a, 5c, Supplementary Data 3). Supporting our autoregulation hypothesis, AcFRS/AzMF-2-F's higher ncAA-dependent RRE fold-increases were achieved by having a substantially lower background RRE in the absence of ncAA while maintaining a high RRE in the presence of ncAA (e.g., 1 mM AzMF) (Figs. 2a, 5c, Supplementary Data 3). Since AcFRS/AzMF-2-F had one synonymous mutation (F236F) in addition to W260L and the TAG mutations, we constructed the W260L.Q345*(TAG) mutant of AcFRS from its wild-type encoding gene and confirmed that the higher ncAA-dependent RRE fold-increases were not due to the F236F mutation (Fig. 5d, e). To exclude the possibility that regulation was through a mechanism independent of amber codon self-read-through, we constructed W260L.Q345*(TAA) and W260L(1-345) mutants of AcFRS, each of which should result in a truncated AcFRS in both the absence and presence of ncAA. We observed no increase in

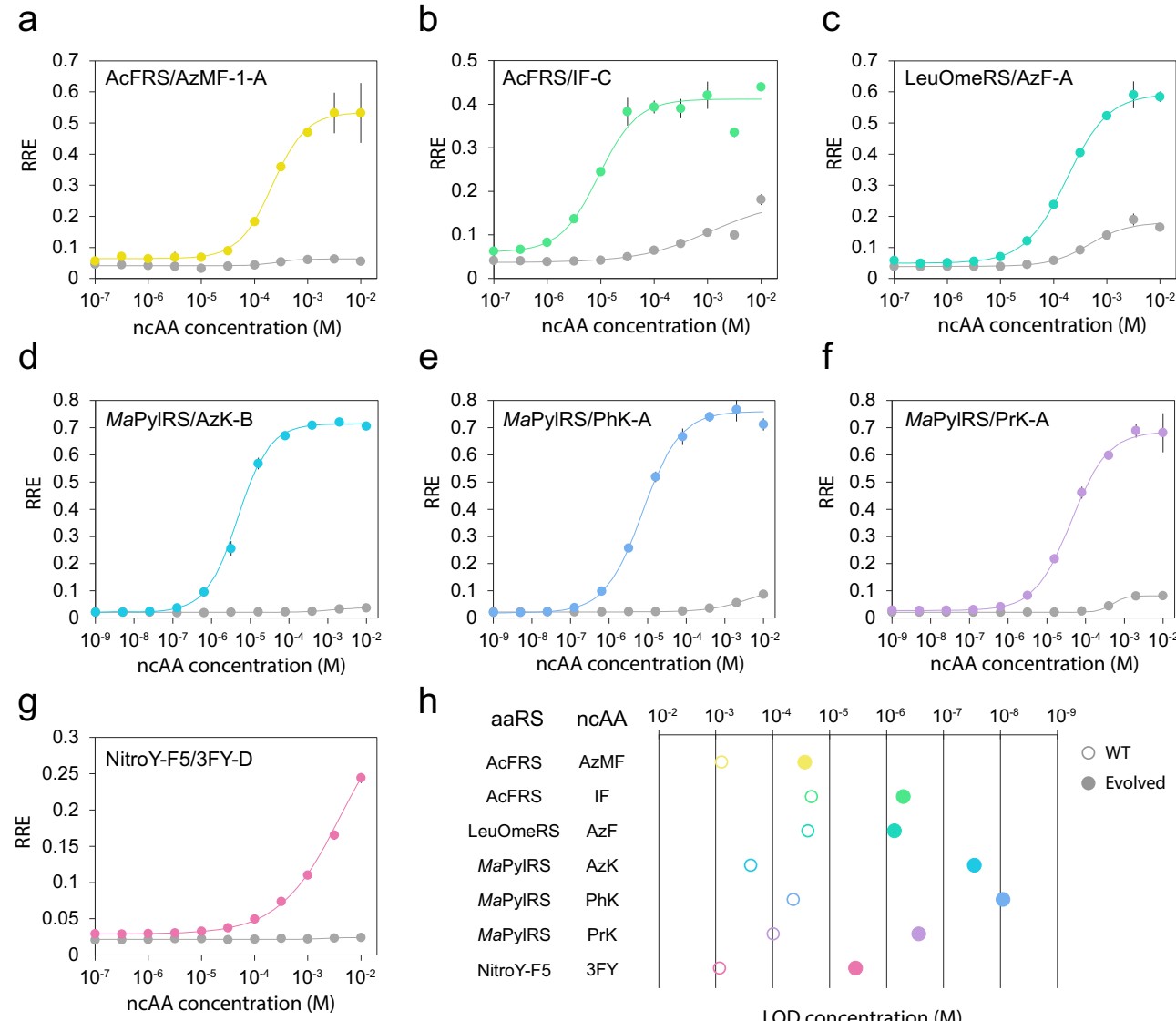

**Fig. 4 | Response function of evolved aaRSs for target ncAAs.** Sensitivity to ncAA concentration for the top clone in each evolution campaign AcFRS/AzMF-1-A (**a**), AcFRS/IF-C (**b**), LeuOmeRS/AzF-A (**c**), *Ma*PylRS/AzK-B (**d**), *Ma*PylRS/PhK-A (**e**), *Ma*PylRS/PrK-A (**f**), and NitroY-F5/3FY-D (**g**). Lines in color correspond to the top performing clones from each evolution campaign and gray lines correspond to the parental aaRSs. Each condition was measured in technical quadruplicates, and the mean ± one standard deviation (error bars) is shown. **h** Sensitivity improvements of aaRSs isolated in each evolution campaign. The limit of detection (LOD) concentration (M), the ncAA concentration which gives an RRE value equivalent to the RRE + 3 S.D. without ncAA, is plotted. Each open circle and filled circle represents a parent aaRS's sensitivity and the corresponding evolved aaRS's sensitivity, respectively. Source data are provided as a Source Data file.

RRE in the presence of AzMF (**5**) and other ncAAs (Fig. 5d, e), consistent with our hypothesized autoregulation mechanism. To test the possibility that mutation of Q345 to AzMF (**5**) improved AcFRS's performance through the chemical identity of AzMF (**5**), we constructed a W260L.Q345F mutant of AcFRS. We found that W260L.Q345F behaved identically to W260L (Fig. 5d, e), suggesting that Q345F did not enhance the activity of AcFRS. To the extent that F is chemically similar to AzMF (**5**), the superior function of AcFRS/AzMF-2-F can likely be assigned to the amber codon at position Q345 rather than the chemical change at Q345 to AzMF (**5**), again consistent with our hypothesis of autoregulation.

In this work, we developed an OrthoRep-driven strategy for aaRS evolution to expand the genetic code of yeast and demonstrated its use in 8 independent evolution campaigns for 7 distinct aaRSs/ncAA combinations, resulting in the generation of a variety of aaRS clones that support the efficient translation of amber codon-containing genes

with at least 13 distinct ncAAs. In contrast to *E. coli*-based continuous evolution platforms[61], our OrthoRep-driven aaRS evolution platform readily admitted the evolution of both bacterial and archaeal aaRSs since it is a yeast-based evolution system where many bacterial aaRS/tRNA pairs are naturally orthogonal. As evidence for the scale and effectiveness of OrthoRep-driven aaRS evolution, (1) all 8 evolution campaigns we attempted were successful, resulting in the ncAA-dependent translation of amber codon-containing genes; (2) translation with ncAAs in some cases reached similar efficiency levels as natural translation with cAAs, where RREs approached 1.0, much higher than RREs achieved previously in yeast GCE efforts; (3) evolved PylRSs also enabled efficient ncAA incorporation in a different organism, *E. coli*; and (4) our experiments yielded a surprising aaRS that regulates its own expression to enforce greater ncAA dependency for the incorporation of ncAAs into proteins. Notably, ncAA-dependent aaRS autoregulation has been deliberately engineered before[62], but

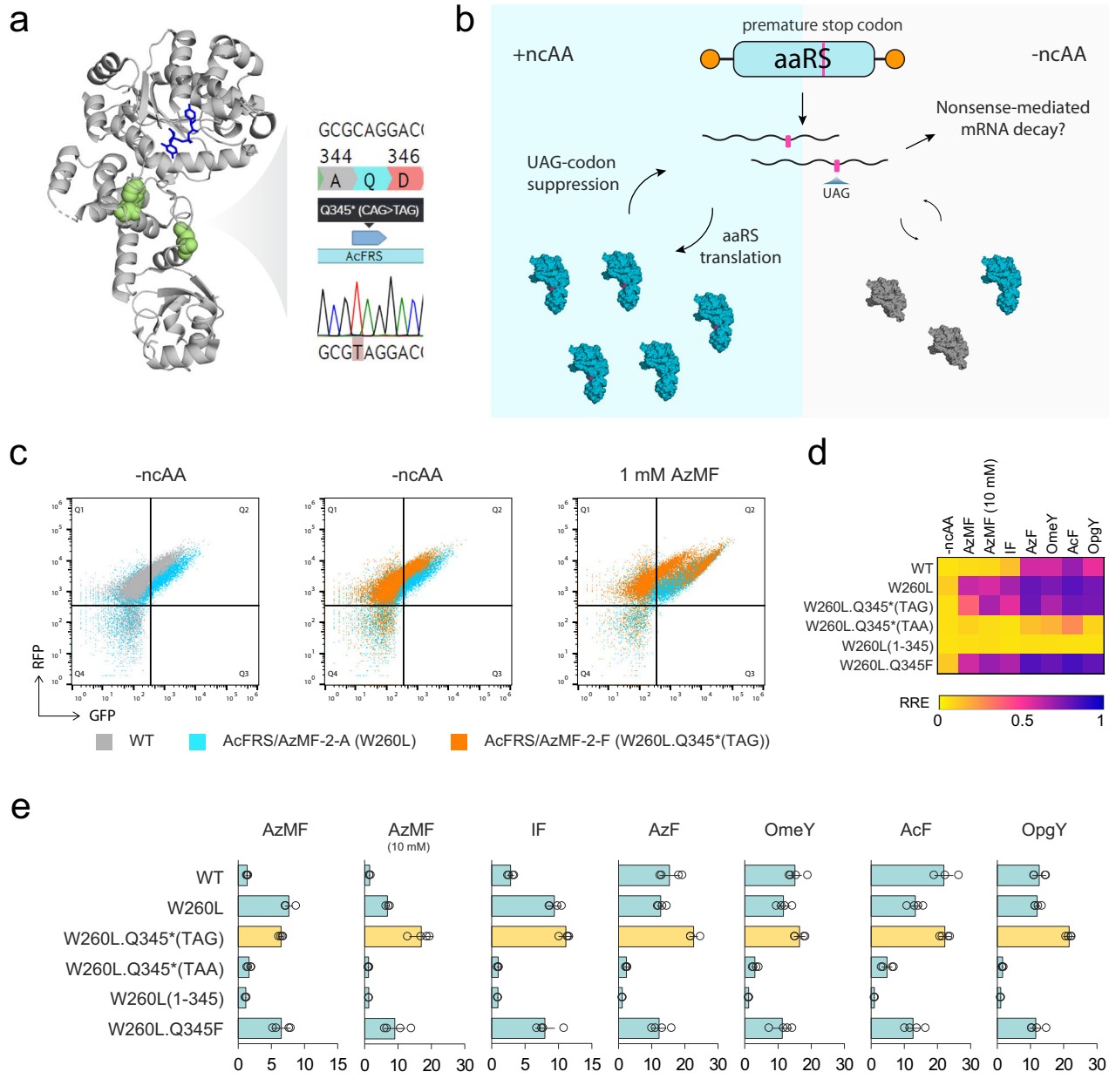

**Fig. 5 | Self-regulation for an aaRS that evolved to contain an amber stop codon. a** Crystal structure of *Ec*TyrRS (PDB 6HB5) with mutated residues in the AcFRS/AzMF-2-F variant highlighted in green. Q345 is mutated to an amber stop codon as shown in the sequencing trace. *Ec*TyrRS is the parent aaRS of AcFRS. **b** Schematic for self-regulation by ncAA-dependent suppression of an amber stop codon within the aaRS. Pink lines, blue enzymes, and gray enzymes indicate UAG stop codons, the full-length aaRS, and the truncated aaRS, respectively. **c** Flow cytometry plots for AcFRS WT (gray), AcFRS/AzMF-2 A (blue), and AcFRS/AzMF-2 F (orange) with -ncAA and 1 mM AzMF. **d** Heatmap of RRE values for AcFRS variants with 6 tyrosine or phenylalanine analogs. **e** Fold change in the RRE of AcFRS variants in the presence and absence of 6 tyrosine or phenylalanine analogs. RRE activity with the ncAA was divided by that without ncAA. Each condition was measured in biological quadruplicates, and the mean ± one standard deviation (error bars) is shown. Source data are provided as a Source Data file.

not via directed evolution, making its independent discovery through our rapid evolution strategy a reassuring result. These successes of OrthoRep-driven aaRS evolution can be used to guide future work. For example, rather than evolving from individual aaRS parents, it should be relatively straightforward to encode aaRS libraries directly onto OrthoRep to maximize the number of parental sequences available from which evolution can commence. This would result in yeast populations that can be used for the end-to-end generation of effective aaRSs achieving GCE with ncAAs well beyond those in the current work.

## Methods

### DNA plasmid construction

Plasmids used in this study are listed in Supplementary Data 5 along with their DNA sequences. All DNA templates for PCR were derived from previous studies or synthesized as gBlocks (IDT). All primers were synthesized by IDT. Amplicons for construction of plasmids were generated using PrimeSTAR GXL DNA Polymerase (Takara Bio). All plasmids were constructed using Gibson Assembly or Golden Gate Assembly and transformed into chemically competent or electrocompetent *E. coli* strain TOP10 (ThermoFisher). Plasmids were

sequence verified by either Sanger sequencing (Azenta) or whole plasmid sequencing (Primordium).

## Reagents
All ncAA stocks were prepared at a final concentration of 50 mM of the L-isomer. Deionized water was added to solid ncAA to approximately 90% of the final volume and the pH was gradually adjusted with a solution of NaOH as needed to dissolve ncAA. The solution was sterile filtered through a 0.2 µm filter. Filtered solutions were stored at −80 °C. The solutions were thawed and stored at 4 °C for up to 8 weeks for experiments. Commercially available key reagent can be found in Supplementary Data 6.

## Yeast strains and media
All yeast strains used in this study are listed in Supplementary Data 7. Yeast were incubated at 30 °C and were typically grown in synthetic complete (SC) growth medium (20 g/L dextrose, 6.7 g/L yeast nitrogen base w/o amino acids (US Biological), appropriate nutrient drop-out mix (US Biological)) or in MSG (L-Glutamic acid monosodium salt) SC growth medium (20 g/L dextrose, 1.72 g/L yeast nitrogen base w/o ammonium sulfate w/o amino acids (US Biological), appropriate nutrient drop-out mix (US Biological), 1 g/L L-Glutamic acid monosodium salt hydrate (ThermoFisher)) minus appropriate nutrients (referred to as -X where X is either the single letter amino acid code for an amino acid nutrient or U for uracil) corresponding to the selectable markers used to maintain plasmids. SCGR media, where glucose in SC media was replaced with 2% galactose and 2% raffinose as sugar source, was used for GAL1 promoter induction. Where selection for the MET15 marker was required, cells were propagated in media lacking both methionine and cysteine. 500 µL liquid yeast cultures in 96-well deep well plates were incubated with shaking at 750 rpm. All other liquid yeast cultures were incubated with shaking at 200 rpm. Media agar plates were made by combining 2× concentrate of molten agar and 2× concentrate of the desired media formulation. Prior to all experiments, cells were grown to saturation in media selecting for maintenance of any plasmids present.

## Yeast transformations
All yeast transformations, including p1 integrations and polymerase replacement integration were performed using frozen competent cells[63]. For p1 integrations and polymerase replacement integrations, 1–5 µg of plasmid was linearized prior to transformation using either ScaI-HF or EcoRI-HF (both NEB), respectively. For CEN/ARS nuclear plasmid transformations, roughly 100–500 ng of plasmid was transformed. Transformants were selected on the appropriate selective media agar plates. MSG SC -LW media agar plates w/ 100 mg/L nourseothricin and 200 mg/L L-canavanine were used for polymerase replacement integrations. Plates were grown at 30 °C for 2 days for nuclear transformations and genomic integrations and for 4 days for p1 integrations. Genetic deletion for TRP1 was performed using CRISPR/Cas9 with a spacer sequence and a linear DNA fragment comprised of two concatenated homology flanks to the TRP1 gene. The TRP1 deletion was confirmed by PCRing and sequencing the locus. When isolating individual clones from genetic deletion and integration transformations, colonies were restreaked onto media agar plates of the same formulation to ensure isolation of cells with the desired genetic change.

All linearized plasmids for p1 integration were integrated over a split LEU2 landing pad to generate the desired p1 constructs[47]. Extraction of genomic DNA (gDNA) and p1/p2 plasmids was performed for the resulting clonal strains as previously described[64]. In brief, 1.5 mL of yeast culture was pelleted, resuspended in 1 mL 0.9% NaCl, pelleted again, and resuspended in 250 µL Zymolyase solution (0.9 M D-Sorbitol, 0.1 M EDTA, 10 U/mL Zymolyase (US Biological)). After 1 h incubation at 37 °C, the suspension was centrifuged, resuspended in

280.5 µL proteinase K solution (250 µl TE (50 mM Tris-HCl (pH 7.5), 20 mM EDTA), 25 µL 10% sodium dodecyl sulfate, 5.5 µl proteinase K stock solution (10 mg/ml proteinase K (Sigma Aldrich)), and incubated at 65 °C for 30 min. Following incubation, 75 µL 5 M potassium acetate was added, and the mixture was incubated on ice for 30 min. Samples were centrifuged, mixed with 700 µL ethanol, and centrifuged again. The pellet was dried, resuspended in 150 µL TE (50 mM Tris-HCl (pH 7.5), 20 mM EDTA), and centrifuged. Supernatant was combined with 8 µL 1 mg/mL ribonuclease A (VWR), incubated at 37 °C for 30 min, then mixed with 150 µL isopropanol. After centrifugation, the pellet was dried and resuspended in 30 µL water. Proper integration was validated by visualizing the recombinant p1 band of the desired size by gel electrophoresis of the extracted and proteinase K treated DNA. Following confirmation of the presence of the recombinant p1, reporter plasmids were transformed and then polymerase replacement integrations were performed. The presence of the recombinant p1 was also confirmed after polymerase replacement integrations and any evolution campaigns.

## FACS-based aaRS evolution and selection
In preparation for each FACS selection step, yeast strains harboring the gene encoding an aaRS on p1, an appropriate reporter plasmid, and the gene encoding an error-prone orthogonal DNAP (BB-Tv)[47] at the CAN1 locus of the host genome, were grown in SC-LW at 30 °C to saturation and then diluted to $OD_{600} = 0.6$ in 2 mL of the same media. The diluted cultures were grown at 30 °C to $OD_{600}$ 1.5–3 (4–7 h) and then induced in 2 mL of SCGR-LW media at $OD_{600} = 0.6$ supplemented with no ncAA or appropriate ncAA to induce RXG reporter expression and amber stop codon suppression. Induced cultures were incubated at 30 °C for 2 days. After culture saturation, cells were harvested and washed with HBSM buffer (20 mM HEPES pH 7.5, 150 mM NaCl, 5 mM maltose). After resuspending in HBSM buffer, FACS was performed with a Sony SH800S cell sorter using a 100 µm Sony Sorting Chip. For positive sorting, cells were grown in the SCGR-LW media containing an appropriate ncAA, whereas for negative sorting, cells were grown in the SCGR-LW media without any ncAA. Roughly 10,000,000 events and 1,000,000 events were measured for positive and negative sorting, respectively, for each evolution condition. Fluorescence was measured using 488 nm laser with 525/50 filter for GFP and 617/30 filter for RFP detection. The top 0.05% of cells positive for both GFP and RFP were recovered for positive sorts while the top 5% of cells positive for RFP and most negative for GFP were recovered for negative sorts. Cells were recovered in 2 mL of SC-LW at 30 °C until saturation and were used for the next round.

After several rounds of growth and selection, aaRS sequences were amplified from extracted p1 plasmids and subcloned into a CEN/ARS plasmid in a library format (over 1,000,000 colonies), preventing further hypermutation of the evolved aaRS sequences. Libraries were transformed into yeast strains with an appropriate reporter plasmid and all the colonies (over 20,000 colonies) were scraped. Cells were grown in SC-LW to saturation and subjected to a final negative sort and positive sort for each library using the same protocol above. Roughly 1,000,000 events and 5,000,000 events were measured for negative and positive sorts, respectively. For negative sorts, the top 5% of cells positive for RFP and most negative for GFP were recovered. For positive sorts, the top 0.1% of cells positive for both GFP and RFP were recovered. After positive sorts, cells were grown on SC-LW media agar plates to isolate individual clones for sequencing and further characterization.

## Flow cytometry
All measurements to characterize the activity of aaRSs were performed by flow cytometry. Cells were grown in SC-LW media to saturation and then diluted to $OD_{600} = 0.6$ in 500 µL of the same media. The diluted cultures were grown to $OD_{600}$ 1.5–3 (4–7 h) and then induced in 200 µL

of SCGR-LW media at $OD_{600} = 0.6$ supplemented with no ncAA or appropriate ncAA to induce the expression of the RXG reporter containing an amber stop codon for suppression. Induced cultures were incubated for 2 days. After culture saturation, cells were diluted into 0.9% NaCl and their fluorescence was measured on an Attune NxT flow cytometer (Life Technologies). The fluorescence of RFP and GFP was measured for 20,000 events, and the mean fluorescence for each population was determined. Flow cytometry data analyses were performed using FlowJo 10.10.0. Autofluorescence of cells was subtracted using the mean fluorescence of uninduced cells grown in SC media. Fold change was calculated by dividing the mean fluorescence in SCGR-LW media with ncAA by the mean fluorescence in SCGR-LW media without ncAA. Relative readthrough efficiency (RRE) was calculated as previously described[65]. Cells transformed with the plasmid encoding the RYG reporter for RRE calculation were induced in the absence of an ncAA. To fit the response curves, a four-parameter logistic regression using the GRG nonlinear solving method was performed with Solver in Microsoft Excel. To calculate the limit of detection (LOD) concentration, the ncAA concentration which gave an RRE value equivalent to the RRE + 3 S.D. without ncAA, was determined using the fitted titration curve. Each value corresponds to the mean of 3 or 4 biological replicates for characterization of stop codon mutant and 4 technical replicates for the other measurements.

### Growth rate assay
Yeast strains harboring a plasmid encoding *Ma*PylRS/PhK-A or an empty vector and an appropriate RXG reporter plasmid were grown in SC-LW media to saturation and then diluted to $OD_{600} = 0.1$ in $200\,\mu L$ of the same media supplemented with no ncAA or appropriate ncAA in 96-well clear-bottom plates. Plates were then sealed with a porous membrane and allowed to incubate with shaking at 30 °C for 24 h, with $OD_{600}$ measurements taken automatically every 1000 seconds using an Infinite 200 PRO plate reader (Tecan). Growth rate was calculated based on logarithmic transformed $OD_{600}$ in a range of exponential growth.

### Protein purification and mass spectrometry
Yeast strains harboring a plasmid encoding an evolved aaRS and an appropriate sfGFP-150TAG reporter plasmid were grown in SC-LW media to saturation and then diluted to $OD_{600} = 0.6$ in 20 mL of the same media. The diluted cultures were grown to $OD_{600}$ 1.5–3 (4–7 h) and then induced in 40 mL of SCGR-LW media at $OD_{600} = 0.6$ supplemented with appropriate ncAA to induce the expression of the sfGFP-150TAG reporter. Induced cultures were incubated for 2 days. After culture saturation, cells were washed with 0.9% NaCl. Proteins were extracted from yeast cells using Y-PER (ThermoFisher) containing cOmplete, EDTA-free protease inhibitor cocktail (MilliporeSigma), purified using HisPur Ni-NTA resin (ThermoFisher) and eluted with elution buffer (20 mM sodium phosphate, 300 mM NaCl, 250 mM imidazole, pH 8.0). Intact proteins were analyzed by LC/MS (ACQUITY UPLC H-class system and Xevo G2-XS QTof, Waters). Proteins were separated using an ACQUITY UPLC BEH Phenyl VanGuard Pre-column (130 Å, 1.7 µm, 2.1 mm × 5 mm, Waters) at 45 °C. The 5-min method used 0.2 mL/min flow rate of a gradient of Buffer A consisting of 0.1% formic acid in water and Buffer B, acetonitrile. The Xevo Z-spray source was operated in positive MS resolution mode, 400–4000 Da with a capillary voltage of 3000 V and a cone voltage of 40 V (NaCsI calibration, Leu-enkephalin lock-mass). Nitrogen was used as the desolvation gas at 350 °C and a total flow of 800 L/h. Total average mass spectra were reconstructed from the charge state ion series using the MaxEnt1 algorithm from MassLynx software (Waters) according to the manufacturer's instructions. To obtain the ion series described, the major peak of the chromatogram was selected for integration before further analysis. The theoretical molecular weights of proteins with ncAAs were calculated by first computing the theoretical molecular weight of wild-type sfGFP and then manually correcting for the theoretical molecular weight of ncAAs.

### Activity measurements in *E. coli*
TOP10 *E. coli* cells were cotransformed with (1) a plasmid expressing *Ma*tRNA$^{Pyl}_{CUA}$ and an evolved PylRS variant, and (2) a plasmid expressing GFP with one amber codon. Transformed cells were grown in 2YT media supplemented with $50\,\mu g/mL$ spectinomycin and $100\,\mu g/mL$ ampicillin at 37 °C. The saturated cultures were diluted at a ratio of 1:100 and grown to $OD_{600} = 0.6$ (2–3 h). The expression of aaRS and GFP reporter was induced in the same media supplemented with 0.02% of L-arabinose and 1 mM of IPTG. The appropriate ncAA was also added at this point. The cultures were incubated for 18 h at 37 °C. Cells from a $25\,\mu L$ culture were resuspended in $200\,\mu L$ of 0.9% NaCl and transferred to a 96-well black clear bottom plate. GFP fluorescence (excitation/emission, 475/515 nm) was measured using an Infinite 200 PRO plate reader (Tecan). Fluorescence signals were normalized by dividing by $OD_{600}$. The resulting fluorescence for each sample was divided by fluorescence of *E. coli* cells expressing wild-type GFP to obtain the relative fluorescence ratio.

### Statistics and reproducibility
Microsoft Excel was used to analyze the data. The number of replicates is provided in the figure legends. No statistical methods were used to predetermine sample size. An outlier, defined as a value five median absolute deviations from the median of the residuals, was excluded from the data shown in Fig. 5d, e. No data were excluded from any other figures.

### Reporting summary
Further information on research design is available in the Nature Portfolio Reporting Summary linked to this article.

## Data availability
Information on the evolution campaigns, mutations of the evolved variants, commercially available key reagent, plasmids, and yeast strains used in this study are available in Supplementary Data. Source data are provided with this paper.

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

## Acknowledgements

This work was funded by NIH R35GM136297 to C.C.L. and a JSPS Overseas Research Fellowship 202260318 to Y.F.

## Author contributions

Y.F., G.R., J.A.V., and C.C.L. designed the experiments. Y.F. conducted the experiments. Y.F. and C.C.L. analyzed the data and wrote the manuscript with input from all authors.

## Competing interests

C.C.L. is a co-founder of K2 Therapeutics, which uses OrthoRep for protein engineering. The remaining authors declare no competing interests.
