## [Transparent Peer Review file · Nature Communications]

Directed evolution of aminoacyl-tRNA synthetases through in vivo hypermutation

Corresponding Author: Professor Chang Liu

Version 0:

Reviewer comments:

Reviewer #1

(Remarks to the Author)

In this manuscript, Furuhashi et al. demonstrate a method for rapidly evolving aminoacyl-tRNA synthetases (aaRSs) for incorporation of non-canonical amino acids into proteins in yeast. Directed evolution of aaRSs was done using the error-prone DNA replication system OrthoRep and a dual fluorescent reporter protein to select for cells that incorporated the ncAA in response to the amber stop codon. The authors demonstrated that they could evolve aaRSs capable of relative read-through efficiencies of >10-fold higher than the starting aaRS in some cases. The authors then went on to further describe the characteristics of the top evolved synthetases, including a case where one of the top evolved aaRSs contained an amber stop codon in the reading frame. The work described by Furuhashi et al. contributes to the field of GCE by reducing the time and effort needed to evolve aaRSs. Below are my general comments about the manuscript, followed by more detailed questions.

The overall flow of the manuscript was logical and easy to read, and the figures and legends were clear. The introduction contained sufficient background information for the reader, and introduced both the problem and solution. I appreciated that the directed evolution strategy was described in enough detail that the work could be readily reproduced. The analysis of evolved aaRSs was thorough, including the unexpected result of an evolved aaRS containing an amber stop codon – which was investigated in additional detail. Finally, the conclusions summarize the work and propose a future directed evolution campaign in yeast containing a library of starting aaRSs, as opposed to a single parent – an interesting idea. I do have some questions and recommendations that I think could make the manuscript even stronger.

1. Often times, when new aaRSs are evolved, mass spectrometry is used to confirm that the ncAA is incorporated in response to the amber stop codon, and not a CAA. In this system, because the amber stop codon is contained in the linker between the two fluorescent proteins, incorporation of any AA will result in increased RRE. You have done a nice job with the -ncAA condition and the negative selection to prevent aminoacylation of CAAs; however, I think that one example of mass spectrometry would strengthen your results even further.

2. Evolved aaRSs that enable efficient incorporation of ncAAs at the desired amber stop codon also enable efficient incorporation of the ncAA at amber stop codons in the genome. Incorporation at genomic UAGs can potentially have a significant effect on the health of the cells, especially if essential genes are terminated by amber stop codons. How is the growth & cell health of the yeast harboring the evolved aaRS/tRNA pairs affected in the presence of the ncAAs? Do they reach densities comparable to a -ncAA condition? Or would you expect a decreased yield of protein containing the ncAA due to growth limitations? I think it would be worthwhile to briefly discuss.

3. Synonymous mutations could potentially improve expression. I appreciate that you included synonymous mutations in your supplementary tables and have an example (in the autoregulation section) of constructing the mutant from the starting parent independently. Do you expect that any of the variants containing a large number of synonymous mutations (for example, LeuOmeRS/AzF-B, Table S3, 6 synonymous mutations) have improved expression? Have you re-constructed any other variants outside of the autoregulation aaRS? It may be worth commenting on the frequency of synonymous mutations generated in your directed evolution campaigns.

4. You describe in your introduction the use of yeast as an ideal organism for directed evolution of aaRSs, because of its compatibility with a wide range of bacterial and archeal pairs. Have you tried any of your evolved aaRS/tRNA pairs in *E. coli*

or mammalian cells to show the broad utility of aaRSs evolved with your yeast system?

5. Very minor formatting note:

a. Line 516 – I think a typo: “mutant of AcFRS from its wild-type encoding gene and confirmed....”

Overall, this work was well-done, will have an impact on the GCE field, and I think it is appropriate for publication in Nature Communications pending answers to the questions above. Thank you for the opportunity to review your manuscript and I look forward to your responses and additional discussions.

Reviewer #2

(Remarks to the Author)

In this work, Furuhashi and colleagues utilize a unique hypermutation platform in conjunction with FACS and a REF-TAG-GFP reporter to perform continuous synthetic evolution on a number of aaRSs in order to identify new aaRSs capable of encoding a number of ncAAs (none of the ncAAs or aaRSs are novel). The authors report that in all selections, success was achieved and thoroughly characterized both the activity and sequence of top-performing hits. In many cases, the authors identify new mutations that have not been uncovered from previous directed evolution efforts. Interestingly, the authors also identify a pair of AcFRSs selected for AzMF that contain internal UAG codon codons, which, through the characterization of several distinct mutants, conclude likely plays an autoregulatory role that decreases background levels of suppression in the absence of AzMF. This platform promises to be a powerful new tool for selecting novel aaRSs, since it, in principle, has the potential to explore sequence space outside of the confines of current directed evolution platforms and can achieve suppression efficiencies at least as high as previous approaches. Additionally, yeast is a valuable model organism and for bioproduction and protein engineering with yeast display. Advances in GCE components for yeast will be of significant value to the field.

Edits

1. Yeast is less used organism for GCE and therefore mass spec analysis of ncAA-proteins should be used to confirm encoding of the indicated ncAAs. This is a standard practice for GCE protein engineering papers, particularly with new aaRS hits.
2. The abstract (lines 47-49) stipulates that the activity of the selected aaRS is “reached efficiencies for amber codon-specified ncAA-dependent translation comparable to translation with natural amino acids specified by sense codons in yeast”
 - a. This misleading as it implies that the selected aaRS have catalytic efficiency comparable to natural aaRS (which support many 1000s of codons), but in reality is supports high suppression efficiency at a single site in a single reporter. This should be changed to be less misleading or tRNA/RS concentrations and catalytic constants should be measured.
3. I think the “design” on line 248 is supposed to say “designated?”
4. The final sentence of the first paragraph in the results section “Characteristics of evolved E. coli aaRSs” (lines 263-265) states that the OrthoRep selections maintained or decreased promiscuity of aaRSs; this is not completely accurate
 - a. For example, in figure 2a for AcFRS/AzMF-1, clone B appears to have lower activity for AcF than the parent (itself an AcFRS)
 - b. This would imply that clone B (and indeed others) seem to become less promiscuous for this ncAA, not more.
 - c. This statement could also be interpreted on a general context, wherein most clones appear to have higher activity with the screened ncAAs, but is this just due to improved catalysis or expanded promiscuity?
5. I find it fascinating that the two AzMF selections differed so substantially from one another. In the second replicate campaign, it appears as if the W260L mutation became fixed early on and subsequently diversified slightly, whereas this mutation never had a chance to arise in the first replicate campaign.
 - a. Can the authors comment on why these replicate yielded such different outcomes regarding hit sequences and diversity?
6. In the caption for Figure 2 (line 344), it is indicated that the mutated residues in the structures are presented as blue “sticks,” but I believe these are technically referred to as “spheres?”
7. In line 353, it is indicated that AcFRS/IF clone C only had the V101A mutation, but it also appears that clone E also had this same sequence?
8. As these ncAAs seems similar in size why did selection for AzK and PhK yield aaRSs that had lower activity for BockK, but selection for PrK yield aaRSs with increased activity for BockK?
9. It would be useful if the authors commented on what role the observed mutations distal to the active site might be having on activity?
10. First line in Figure 4 caption (line 553) should say “(a-g)” instead of “(a-f)”
11. For the autoregulation hypothesis, it must be true that, at least initially, the TAG codon of the putatively-autoregulated aaRSs must be readthrough by natural near-cognate suppression (NCS). What amino acids are commonly encoded via NCS in response to UAG codons in yeast? Are these “1st generation” aaRSs weaker or more active than the AzMF mutant?
12. regarding the autoregulation, could the authors comment on when, explicitly, the negative selection was performed during the AzMF-2 selection?

Version 1:

Reviewer comments:

Reviewer #1

(Remarks to the Author)

Thank you to the authors for thorough, thoughtful responses and for providing additional data. The authors have addressed all of my questions and recommendations, and I recommend this article for publication.

Reviewer #2

(Remarks to the Author)

The authors have adequately address the reviewers concerns.

Reviewer #1 (Remarks to the Author):

In this manuscript, Furuhashi et al. demonstrate a method for rapidly evolving aminoacyl-tRNA synthetases (aaRSs) for incorporation of non-canonical amino acids into proteins in yeast. Directed evolution of aaRSs was done using the error-prone DNA replication system OrthoRep and a dual fluorescent reporter protein to select for cells that incorporated the ncAA in response to the amber stop codon. The authors demonstrated that they could evolve aaRSs capable of relative read-through efficiencies of >10-fold higher than the starting aaRS in some cases. The authors then went on to further describe the characteristics of the top evolved synthetases, including a case where one of the top evolved aaRSs contained an amber stop codon in the reading frame. The work described by Furuhashi et al. contributes to the field of GCE by reducing the time and effort needed to evolve aaRSs. Below are my general comments about the manuscript, followed by more detailed questions.

The overall flow of the manuscript was logical and easy to read, and the figures and legends were clear. The introduction contained sufficient background information for the reader, and introduced both the problem and solution. I appreciated that the directed evolution strategy was described in enough detail that the work could be readily reproduced. The analysis of evolved aaRSs was thorough, including the unexpected result of an evolved aaRS containing an amber stop codon – which was investigated in additional detail. Finally, the conclusions summarize the work and propose a future directed evolution campaign in yeast containing a library of starting aaRSs, as opposed to a single parent – an interesting idea. I do have some questions and recommendations that I think could make the manuscript even stronger.

We thank the reviewer for their thorough consideration of our work and insightful comments.

1. Often times, when new aaRSs are evolved, mass spectrometry is used to confirm that the ncAA is incorporated in response to the amber stop codon, and not a CAA. In this system, because the amber stop codon is contained in the linker between the two fluorescent proteins, incorporation of any AA will result in increased RRE. You have done a nice job with the -ncAA condition and the negative selection to prevent aminoacylation of CAAs; however, I think that one example of mass spectrometry would strengthen your results even further.

Thank you for your valuable suggestion. We have now expressed and purified sfGFP containing an amber stop codon at position 150 in the presence of our best performing variants and corresponding ncAAs. Mass spectrometry of the sfGFP-150TAG proteins showed expected masses for 6 out of 7 best performing variants, indicating ncAA incorporation at the position 150, not a cAA (Supplementary Figure 2). We could not obtain enough sfGFP-150TAG protein for NitroY-F5/3FY-D due to its relatively low RRE with 3FY and the toxicity of 3FY at 10 mM. We believe that these results strengthen the effectiveness of our OrthoRep-driven aaRS evolution system.

2. Evolved aaRSs that enable efficient incorporation of ncAAs at the desired amber stop codon also enable efficient incorporation of the ncAA at amber stop codons in the genome. Incorporation at genomic UAGs can potentially have a significant effect on the health of the cells, especially if essential genes are terminated by amber stop codons. How is the growth & cell health of the yeast harboring the evolved aaRS/tRNA pairs affected in the presence of the ncAAs? Do they reach densities comparable to a -ncAA condition? Or would you expect a decreased yield of protein containing the ncAA due to growth limitations? I think it would be worthwhile to briefly discuss.

Thank you for this helpful suggestion. We agree that incorporation at genomic UAGs can potentially influence the health of the cells. To evaluate the effect of genomic UAG suppression on cell growth, we have now performed growth rate assays of a yeast strain expressing MaPylRS/PhK-A, which exhibited the highest RRE with 1 mM PhK. While the strain showed slightly slower growth (growth rate of 0.243 per hour) with 1 mM PhK compared to the same strain in -ncAA condition (growth rate of 0.260 per hour) and a control strain harboring an empty vector with 1 mM PhK (growth rate of 0.257 per hour), it grew and saturated normally (Supplementary Figure 1). Thus, we believe that genomic UAG codon suppression does not have a critical effect on cell yeast growth. This is now described and explained in the main text.

3. Synonymous mutations could potentially improve expression. I appreciate that you included synonymous mutations in your supplementary tables and have an example (in the autoregulation section) of constructing the mutant from the starting parent independently. Do you expect that any of the variants containing a large number of synonymous mutations (for example, LeuOmeRS/AzF-B, Table S3, 6 synonymous mutations) have improved expression? Have you re-constructed any other variants outside of the autoregulation aaRS? It may be worth commenting on the frequency of synonymous mutations generated in your directed evolution campaigns.

We agree that synonymous mutations could improve expression. Although we have not reconstructed any of our evolved variants except the autoregulation aaRS, it is possible that we enriched variants with improved expression especially during post-evolution selection, where we used lower expression setup compared to that during evolution. However, we think our evolution campaigns were mostly driven by nonsynonymous mutations due to the fact that the overall nonsynonymous to synonymous mutation ratio of all mutations was higher than a null model, suggesting adaptation was overall driven by nonsynonymous mutations. However, we have not done a detailed analysis of this as we do not see it to be critical to the main claims, and therefore refrain from commenting.

4. You describe in your introduction the use of yeast as an ideal organism for directed evolution of aaRSs, because of its compatibility with a wide range of bacterial and archeal pairs. Have you tried any of your evolved aaRS/tRNA pairs in *E. coli* or mammalian cells to show the broad utility of aaRSs evolved with your yeast system?

*Thank you for this helpful suggestion. To support this claim, we have included new experiments that test the activity of four evolved PylRSs in *E. coli* using a GFP reporter with an amber codon inserted between position 2 and 3. They all showed consistent activity as that found through RRE measurements in yeast with the exception of MaPylRS/PrK-A, which showed unexpectedly low activity with 1 mM PrK (Supplementary Figure 3). This could be caused by the difference in intracellular PrK concentration and the low response of MaPylRS/PrK-A to PrK, as we found much higher activity with 10 mM PrK in *E. coli*. This experiment and associated discussions are included in the revision.*

5. Very minor formatting note:

a. Line 516 – I think a typo: “mutant of AcFRS from its wild-type encoding gene and confirmed....”

We have revised the sentence.

Overall, this work was well-done, will have an impact on the GCE field, and I think it is appropriate for publication in Nature Communications pending answers to the questions above. Thank you for the opportunity to review your manuscript and I look forward to your responses and additional discussions.

Thank you!

Reviewer #2 (Remarks to the Author):

In this work, Furuhashi and colleagues utilize a unique hypermutation platform in conjunction with FACS and a REF-TAG-GFP reporter to perform continuous synthetic evolution on a number of aaRSs in order to identify new aaRSs capable of encoding a number of ncAAs (none of the ncAAs or aaRSs are novel). The authors report that in all selections, success was achieved and thoroughly characterized both the activity and sequence of top-performing hits. In many cases, the authors identify new mutations that have not been uncovered from previous directed evolution efforts. Interestingly, the authors also identify a pair of AcFRSs selected for AzMF that contain internal UAG codon codons, which, through the characterization of several distinct mutants, conclude likely plays an autoregulatory role that decreases background levels of suppression in the absence of AzMF. This platform promises to be a powerful new tool for selecting novel aaRSs, since it, in principle, has the potential to explore sequence space outside of the confines of current directed evolution platforms and can achieve suppression efficiencies at least as high as previous approaches. Additionally, yeast is a valuable model organism and for bioproduction and protein engineering with yeast display. Advances in GCE components for yeast will be of significant value to the field.

We thank the reviewer for their thorough consideration of our work and kind comments.

Edits

1. Yeast is less used organism for GCE and therefore mass spec analysis of ncAA-proteins should be used to confirm encoding of the indicated ncAAs. This is a standard practice for GCE protein engineering papers, particularly with new aaRS hits.

Thank you for your valuable suggestion. We expressed and purified sfGFP containing an amber stop codon at position 150 in the presence of our best performing variants and corresponding ncAAs. Mass spectrometry of the sfGFP-150TAG proteins showed expected masses for 6 out of 7 best performing variants, indicating ncAA incorporation at the position 150, not a cAA (Supplementary Figure 2). We could not obtain enough sfGFP-150TAG protein for NitroY-F5/3FY-D due to its relatively low RRE with 3FY and the toxicity of 3FY at 10 mM. We believe that these results strengthen the effectiveness of our OrthoRep-driven aaRS evolution system.

2. The abstract (lines 47-49) stipulates that the activity of the selected aaRS is “reached efficiencies for amber codon-specified ncAA-dependent translation comparable to translation with natural amino acids specified by sense codons in yeast”

a. This misleading as it implies that the selected aaRS have catalytic efficiency comparable to natural aaRS (which support many 1000s of codons), but in reality is supports high suppression efficiency at a single site in a single reporter. This should be changed to be less misleading or tRNA/RS concentrations and catalytic constants should be measured.

We revised the sentence to clarify this point. Thank you.

3. I think the “design” on line 248 is supposed to say “designated?”

Thank you for pointing this out. We revised it.

4. The final sentence of the first paragraph in the results section “Characteristics of evolved E. coli aaRSs” (lines 263-265) states that the OrthoRep selections maintained or decreased promiscuity of aaRSs; this is not completely accurate

a. For example, in figure 2a for AcFRS/AzMF-1, clone B appears to have lower activity for AcF than the parent (itself an AcFRS)

b. This would imply that clone B (and indeed others) seem to become less promiscuous for this ncAA, not more.

c. This statement could also be interpreted on a general context, wherein most clones appear to have higher activity with the screened ncAAs, but is this just due to improved catalysis or expanded promiscuity?

Thank you, we have revised to clarify this point.

5. I find it fascinating that the two AzMF selections differed so substantially from one another. In the second replicate campaign, it appears as if the W260L mutation became fixed early on and subsequently diversified slightly, whereas this mutation never had a chance to arise in the first replicate campaign.

a. Can the authors comment on why these replicate yielded such different outcomes regarding hit sequences and diversity?

As we picked different colonies for the two AzMF evolution campaigns after epDNAP integration, the individual colonies accumulated different mutations on AcFRS as they grew before starting the evolution campaigns. Since there are at least 3, and potentially more ways to get AzMF activity, they presumably chose the fastest way to solve the selection pressure using the given set of mutations in each evolution campaign. We think that is why these replicates yielded such different outcomes.

6. In the caption for Figure 2 (line 344), it is indicated that the mutated residues in the structures are presented as blue “sticks,” but I believe these are technically referred to as “spheres?”

The blue sticks indicate the amino acid substrates of aaRSs, not the mutated residues.

7. In line 353, it is indicated that AcFRS/IF clone C only had the V101A mutation, but it also appears that clone E also had this same sequence?

Yes, that is true. We revised the sentence to clarify this point.

8. As these ncAAs seems similar in size why did selection for AzK and PhK yield aaRSs that had lower activity for Bock, but selection for PrK yield aaRSs with increased activity for Bock?

We do not have a good explanation on this. Since AzK and PhK have a couple of nitrogen atoms in their side chain but Bock and PrK do not, we suspect MaPylRS evolved to interact with those nitrogen atoms in evolution campaigns for AzK and PhK, which caused the decreased activity for Bock. But we do not know.

9. It would be useful if the authors commented on what role the observed mutations distal to the active site might be having on activity?

Thank you for your suggestion. In MaPylRS and NitroY-F5 evolution campaigns, we observed a few shared mutations in their tRNA binding domain including G17, N18, and A43. We think these mutations contribute on improving the interaction with tRNA. We discussed this point in the revised manuscript.

10. First line in Figure 4 caption (line 553) should say "(a-g)" instead of "(a-f)"

Thank you for pointing this out. We revised it.

11. For the autoregulation hypothesis, it must be true that, at least initially, the TAG codon of the putatively-autoregulated aaRSs must be readthrough by natural near-cognate suppression (NCS). What amino acids are commonly encoded via NCS in response to UAG codons in yeast? Are these "1st generation" aaRSs weaker or more active than the AzMF mutant?

Thank you for pointing this out. We agree that the UAG codon of the autoregulated aaRSs must be readthrough by natural near-cognate suppression initially. The most common amino acid incorporated by natural near-cognate suppression for UAG codons is tyrosine in yeast, according to the previous study (doi: 10.1093/nar/gku663). Since the Q345 is permissive for amino acid substitution at least for phenylalanine and tyrosine analogs as we showed in Figure 5d, we assume the "1st generation" aaRS would have the similar activity to the AzMF mutant.

12. regarding the autoregulation, could the authors comment on when, explicitly, the negative selection was performed during the AzMF-2 selection?

The negative selection was performed at round 2 and 5 for the AzMF-2 evolution campaign. We included this in Supplementary table 1.